# "When the pain is so acute or if I think that I'm going to die": Health care seeking behaviors and experiences of transgender and gender diverse people in an urban area

Mandi L. Pratt-Chapman[1]*, Jeanne Murphy[2,3], Dana Hines[4], Ruta Brazinskaite[1], Allison R. Warren[5,6], Asa Radix[7]

1 GW Cancer Center, School of Medicine and Health Sciences, The George Washington University, Washington, DC, United States of America, 2 GW Cancer Center, School of Nursing, The George Washington University, Washington, DC, United States of America, 3 The GW Medical Faculty Associates, The George Washington University, Washington, DC, United States of America, 4 Division of Community HIV/AIDS Programs, HIV/AIDS Bureau, Rockville, MD, United States of America, 5 PRIME Center of Innovation, VA Connecticut Healthcare System, West Haven, CT, United States of America, 6 Department of Psychiatry, Yale School of Medicine, New Haven, CT, United States of America, 7 Callen-Lorde Community Health Center, New York, NY, United States of America

* mandi@gwu.edu

**Data Availability Statement:** The GW IRB determined that data cannot be shared publicly,

## Abstract

### Introduction

Approximately 1.4 million transgender and gender diverse (TGD) adults in the United States have unique health and health care needs, including anatomy-driven cancer screening. This study explored the general healthcare experiences of TGD people in the Washington, DC area, and cancer screening experiences in particular.

### Methods

Twenty-one TGD people were recruited through word of mouth and Lesbian Gay Bisexual Transgender Queer (LGBTQ)-specific community events. Participant interviews were conducted and recorded via WebEx (n = 20; one interview failed to record). Interviews were transcribed using Rev.com. Two coders conducted line-by-line coding for emergent themes in NVivo 12, developed a codebook by consensus, and refined the codebook throughout the coding process. Member checking was conducted to ensure credibility of findings.

### Results

Three major themes served as parent nodes: health-care seeking behaviors, quality care, and TGD-specific health care experiences. Within these parent nodes there were 14 child nodes and 4 grand-child nodes. Subthemes for health care seeking behaviors included coverage and costs of care, convenience, trust/mistrust of provider, and provider recommendations for screening. Subthemes for quality of care included professionalism, clinical competence in transgender care, care coordination, provider communication, and patient

because the informed consent indicated that data would only be viewed by the study team. Ethical concerns are related to the sensitivity of the topics discussed as well as sex, gender, geography and age identifiers that increase the likelihood of violating participant anonymity. Questions regarding this decision or data inquiries may be sent to to the GW IRB at ohrirb@gwu.edu with cc: mandi@gwu.edu.

**Funding:** This study was supported by an unrestricted grant from the Avon Foundation to lead author MPC. The funders had no role in study design, data collection and analysis, decision to publish, or preparation of the manuscript. Funder site: https://www.avonworldwide.com/supporting-women/avon-foundation-for-women.

**Competing interests:** No authors have competing interests.

self-advocacy. Overall, transgender men were less satisfied with care than transgender women.

## Conclusions

Results suggest a need for improved provider communication skills, including clear explanations of procedures and recommendations for appropriate screenings to TGD patients. Results also suggest a need for improved clinical knowledge and cultural competency. Respondents also wanted better care coordination and insurance navigation. Overall, these findings can inform health care improvements for TGD people.

## Introduction

Transgender and gender diverse (TGD) people are described as individuals with gender identity different from sex assigned at birth [1, 2]. The Williams Institute estimates that in 2016, approximate 1.4 million adults identified as TGD in the United States or 0.6% of the total population [3]. This subpopulation is particularly susceptible to health care discrimination, including harassment and denial of service [4–6]. Financial challenges, such as lack of insurance coverage for procedures related to the individual's gender identity, may also impede access to health care [7, 8]. For example, according to the results of the U.S. Transgender Survey (USTS) in 2015, approximately 33% of TGD patients postponed seeing a provider due to cost, and 14% of TGD individuals had no insurance compared to 9.1% of the general population [6, 9]. Lack of a primary care provider poses an additional challenge for some TGD people. In the USTS, 15% of respondents reported that they had no routine health provider or hormone provider. One-third of respondents reported at least one negative experience in the last year, including denial of care, verbal harassment or physical abuse [6, 10].

Variability in the anatomy of TGD bodies, use of gender-affirming hormones, and surgical interventions make determination of appropriate cancer screening programs a challenge for health care providers [10–12]. For example, a potential risk factor for cancer in the TGD population is gender-affirming hormonal treatment (HT), but the long-term impact of HT on health outcomes for TGD people is not clear, and not all TGD people elect to have HT. Many TGD individuals elect to have top surgery (breast augmentation or removal with male contouring) but not bottom surgery (e.g., vaginoplasty or phalloplasty) [6, 13]; and some TGD people do not have any surgical interventions. These variations in HT and body modifications affect risk for various types of cancer and impact the types of cancer screenings that are warranted [11, 14, 15].

Yet current nationally-recognized standards for cancer screening are not helpful in guiding primary care providers in screening recommendations for TGD patients. For example, the U. S. Preventive Services Task Force recommendations for cervical cancer screening refer solely to women with a cervix, which neglects to emphasize the need for screening among TGD people with a cervix [7, 16, 17]. The lack of medical guidelines coupled with TGD-specific barriers to care discourage screening among TGD people [10]. Gender dysphoria can be a significant source of distress and further discourages uptake of certain cancer screenings such as cervical cancer screening in transgender men [18]. Gender dysphoria manifests as persistent feelings of identification with a gender different from the gender and sex to which one is assigned [2]. According to the 2015 National Transgender Survey Report, only 27% of transgender men with a cervix reported receiving a Papanicolau (Pap) test to screen for cervical cancer [19] and

even within LGBTQ-affirming health settings transgender men are less likely to receive Pap tests compared with cisgender women [20]. Pap tests have also been shown to be unreliable for cervical cancer screening among transgender men who use testosterone [21].

Transgender women also face cancer screening challenges. Although most transgender women do not have cervices, those who have undergone vaginoplasty using penile and scrotal skin are at risk of developing HPV related lesions [22, 23]. However, lack of research in this population makes it difficult to know whether to recommend screening using cytology [8]. Breast cancer prevention for transgender women is also complicated due to the unclear level of risk. For example, a study in the Netherlands showed that transgender women have an increased risk of developing breast cancer in comparison to cisgender men but a lower risk than cisgender women [15]. The University of California at San Francisco's Center for Excellence for Transgender Health recommends that transgender women age 50 or over with at least 5 years of estrogen exposure start mammography biennially [24]. The prevalence of mammography among transgender women is low, with only 55% of transgender women up to date with screening recommendations compared to 70% of cisgender women [10]. Transgender men may continue to be at risk from breast cancer, even after having top surgery, or removal of the breasts [12]. Colorectal cancer (CRC) screening, which is a non-sex-based screening and not affected by gender-affirming interventions, provides a suitable screening comparison of screening behaviors between cisgender and TGD individuals. Among transgender people the rates of CRC are lowest among transgender women and gender non-binary individuals while transgender men have similar rates of screening to their cisgender counterparts [10].

The purpose of this study was to examine the cancer screening experiences of TGD individuals in the Washington, DC metropolitan area in order to inform health care improvements. The research questions were: What are the healthcare experiences of TGD people in the Washington, DC area in general and for cancer screening in particular?

## Materials and methods

### Study participants

**Recruitment.** Participants were recruited by word of mouth and by research study flyers distributed at various TGD community events. Main inclusion criteria were 1) self-identification as TGD and 2) age 40 or older. The age requirement was intended to balance the age ranges of screening onset for various cancers using the US Preventive Task Force and the American Cancer Society Recommendations for cervical, breast and colorectal cancer screening in cisgender individuals [17, 25–29]. Prospective participants were screened for eligibility. Eligible individuals were then scheduled for phone interviews. Respondents were provided with information about the purpose of the study (to obtain the experiences of TGD individuals with cancer screening) as well as the risks and benefits of participation. Participants were informed that interviews would be recorded and that recordings would be shared only with members of the research team for data analysis and discussion. Participants were further advised of the voluntary nature of the study and informed that they could discontinue the interview at any time. Consent was assumed if participants agreed to proceed with the recorded interview. One participant was found to be under the age of 40 after completion of the interview. This transcript was retained and analyzed in the study.

### Data collection

The PI (MPC) developed an interview guide in collaboration with the research team and a dedicated LGBTQI Advisory Board. Three members of the research team (MPC, DH, JM) performed 21 open-ended, in-depth interviews that took between 45 minutes and 1 hour (one

recording lost due to technical difficulties). The interviews were digitally recorded through the WebEx platform, transcribed verbatim using Rev.com [30], and identifiers were redacted. No follow up interviews were performed.

After completing the interviews, participants received a $125 Amazon gift card. Given the equal valuation of participant time and the community-driven approach of the study, this incentive amount was selected to honor the commitment of TGD individuals' time at a rate equal to that of healthcare providers also interviewed for the study. The interviews were conducted between May and August 2019.

## Data analysis

Data were analyzed with NVivo 12.0 (QSR International, Pty, Doncaster, Victoria, Australia) [31]. Two coders (MPC and JM) conducted line by line coding, using an emergent analytic approach. Nodes were combined to identify major themes rather than granular patient-specific experiences. The coders developed a code book which contained the agreed upon main categories and their definitions that served as a guide to verify the congruency of codes across the interviews.

This study was in accordance with international ethical standards and was approved by The George Washington University Institutional Review Board (IRB) (#NCR191213).

## Results

### Sample characteristics

The recruited sample consisted of 21 participants, however, due to technical difficulties one interview was not recorded; therefore, the total sample for analysis was n = 20. Respondents described themselves as transgender male (n = 4), genderqueer/non-binary (n = 2), transgender female (n = 11), and intersex/assigned male at birth/transgender female (n = 1). Two participants indicated they were assigned male at birth, but did not indicate their gender identity beyond confirming they identified as TGD (n = 2). The average age of the participants was 50.94 years old (SD = 8.69). See Table 1 for participant characteristics.

### Themes

Three overarching themes were found: factors for healthcare seeking behaviors, factors for quality healthcare as defined by TGD people, and transgender-specific considerations in the healthcare experience. See Table 2 for a summary of major themes and subthemes.

Respondents described four factors that facilitated or impeded their healthcare seeking behaviors. First, insurance coverage served as a facilitator for healthcare seeking: One transgender woman (age not reported) said, "I like the price, there's zero copay, lab work is free, and so you can run tests without having to bounce your wallet against your knowledge of your

**Table 1. Sample characteristics.**

| Sex assigned at birth | Gender identity | N |
|---|---|---|
| Female | Male | 4 |
| Female | Genderqueer/ non-binary | 2 |
| Male | Female | 11 |
| Male | Did not disclose | 2 |
| Intersex/Male | Female | 1 |
| **TOTAL** | | 20 |

**Table 2. Major themes and subthemes.**

| Theme (Parent Node) | Sub-themes (Child Node) | Grand-child Node |
|---|---|---|
| Healthcare seeking behaviors | Cost/ Insurance coverage | |
| | Convenience | |
| | Trust in provider | |
| | Provider recommendation for screening | |
| | Patient self-advocacy | |
| Quality care | Professionalism | |
| | Professional competence in transgender care | |
| | Care coordination | |
| | Provider communication skills | Clear explanations |
| | | Active listening |
| | | Affirming language |
| Unique experiences due to being TGD | Gender dysphoria | Gender dysphoria specific to cancer screening |
| | Managing gatekeeping | |
| | Misgendering | |
| | Fear of discrimination | |
| | Confusion about cancer screenings | |

own health." Conversely, out of pocket costs were a deterrent to healthcare seeking: "I'm concerned about this assault on people who are less fortunate and who are vulnerable, who don't and can't afford bills. Even those in the middle class who are beginning to struggle with healthcare. I'm concerned about the cost" (Transgender woman, age 62).

A second theme that emerged was convenience. One transgender woman (age 61) described the relative ease of stool-based colorectal cancer screening compared to colonoscopy: "The only thing now is your doctor recommend you go to your doctor, your doctor refer you to, I guess this company, and they send you the kit and you defecate in it, and then after that, you send it back and they'll give you your results. It's so easy." In contrast, another transgender woman (age 62) said that the preparation for a colonoscopy was a barrier to getting screened: "The colonoscopy itself, it doesn't bother me, it's the having to ingest that awful drink."

Trust in the healthcare provider rendering services was another important factor in healthcare seeking. A transgender woman (age 53) said: "Well, I'm always open to my doctor. He's always open to me. So that was maybe make him a good doctor, because I can tell him anything. I can tell him anything about me what's going on with me." In contrast a transgender man (age 45) suggested hidden motives on the part of his provider, signaling distrust: "[T]hat's why the surgery is so expensive, there's only a few specialists who know how to do it. That specialist doesn't always tell the other doctors what they're doing or what they know and I think that's how they guarantee they're specialists, their bread and butter, right?"

Finally, a critical factor in seeking out cancer screening was receiving a provider recommendation. A transgender man (age 54) shared an experience of his doctors being proactive: "I was concerned about oral cancer... the dentist brought that up. Mostly it's because doctors bring it up and then I'll think, Okay, yeah, that's a good idea. You should check me for that. So they haven't found anything but... I guess they did that screening at the dentist." Similarly, a transgender woman (age 53) reported adhering to a colonoscopy recommended by her provider: "When I turned 50 years old, my doctor said, 'When you turn 50,' that was three years ago, she said, 'It's time for you to get your colonoscopy, because you're 50 years old.' I said,

'Okay.' He just made an appointment, and I went to my appointment. That was it." In contrast, another transgender woman (age not reported) said: "I'm not hearing anything, like nobody's giving me any information on [cancer screening]. And so, it's kind of a big point that this is an area that needs examining."

For quality of care, five factors emerged as critical: four were provider-specific and one was patient-specific. Participants indicated that professionalism, professional competence in transgender care, care coordination, and provider communication skills—such as clear explanations, active listening, and affirming language—impacted the quality of their healthcare. A transgender woman (age 62) voiced appreciation for the cleanliness and respectful interactions she saw in her healthcare provider's clinic: "It's always clean. The people, I mean, although they don't do any major surgeries or anything like that, but... you know, it's just a really nice place... I'm comfortable even with other patients sitting in the waiting room. Everybody's just very nice and mannerable [*sic*.] and social, you know?" This combination of a comfortable environment and respectful staff are hallmarks of professionalism. In contrast, a transgender woman (age 45) was almost denied health care and faced microaggressions when calling into her clinic:

> The time I did not like my care was actually when I had... it was during the time, right in the beginning when insurance was paying for trans surgeries. I was supposed to get my implants done at [institution], then they decided they wouldn't work on trans people. It was some kind of... thing. The doctor still wanted... the doctor was fuming. And he still wanted to do me... He was going to do me at [another institution] because he had operating room rights. Then it turned out that they were going to allow me in to do it. But when I called to do the check in... pre-whatever, [the woman on the phone] said, "Well don't wear your stage make up."

These contrasting experiences show participant appreciation for being treated with respect and dignity as patients deserving of quality healthcare.

In addition to professionalism, the clinical competence of the provider emerged as important for quality healthcare, specifically in terms of provider knowledge of transgender-specific healthcare. A transgender woman (age 62) voiced appreciation of her provider's competence:

> Well that's probably the biggest thing, you know, is that the doctors are so very attentive, and they know, you know, when they're checking me out. Maybe there's not something that I've had to tell them, but they're questioning because they're so observant of me as an individual as to what's going on in my life or what's going on with me physically that they are able to detect certain things that maybe question, "Oh, you didn't have this lump here last time you were here. What's going on?" Those are the things, you know. "[Name], you've lost weight." Yeah, so those are the very things that are so very important to me is that you don't question things that you see are the obvious, whether they may not be obvious to me.

In contrast, a transgender man (age 45) indicated that he had to be his own advocate in every healthcare encounter. He recounted a negative experience after having gender-affirming top surgery:

> They have to know what a trans person is so they know what they're seeing when I sit down. I shouldn't have to come out and basically every trans person becomes their own expert because more often than not when we sit down, we have to do a lot of front end explaining to that doctor educating them. Like for example, after I had my top surgery, I

had to actually go into the emergency room. So while I was there, I had to expose myself, and the doctor didn't know what he was seeing so I had to explain to him what the surgery was, what happened, and what he's looking for.

This participant's personal healthcare experiences led him to conclude that "every trans person has to be their own primary care provider because nine times out of 10, the doctor that they're facing has no idea what they're seeing."

In addition, participants voiced a need for patient-centered provider communication, including active listening, affirming language, and clear explanations about their healthcare. One participant who did not report their gender identity or age but disclosed they were of transgender experience described their strong rapport with their healthcare team:

[The doctor and I] are excellent together. We laugh, we talk. The doctor and his assistant, we have a good rapport with each other. We really laugh when we are doing the procedure, we communicate and we are laughing and talking about different stuff. That's very, very good.

Another gender nonconforming (age 59) respondent recounted an excruciating experience during a Pap exam, a direct result of their provider not listening to them.

I screamed very, very loudly. I usually have a little bit of pain when they do a pap exam. But, again, I bear it out of, a doctor who one time told me, it's a little. . . just the way I was. . . my anatomy. He said, "I'm sorry. This hurts when I do this. I'm sorry."But, she kept trying to do it, and I finally said, "Stop. Stop." I'm sure people in the office could hear me. She finally said, "Well, I'm in, so you want me to finish the exam?"

This narrative describes a previous healthcare provider who had helpful communication skills, informing the patient of anatomical considerations and showing empathy, compared to the current provider who physically hurt the patient and then asked if she should finish the exam despite the patient's cries of pain.

Participants also highlighted the need for affirming language of transgender patients. A transgender woman (age 64) said:

Well one of the things I think is very pronounced is that whenever a provider, someone working for a provider would ask you, "How would you like for me refer to you? What are your pronouns?" You know anything that would try to get a clear understanding about how I want to be addressed.

This transgender woman appreciated having her provider directly ask her about her pronouns and how she would like to be referred. Conversely, a transgender man (age 45) explained his negative experiences in breast clinics:

When you go in to the breast clinic, they don't see you at all, they ignore all trans masculine identity. As a matter of fact, I was so upset that when I got upset when I was in there, they thought that using my name was causing me upset, so they went back to the name on record thinking that that would calm me down. They refused whatsoever, because if they're looking at an XX body part, they will treat you with XX, period. The whole identity of a breast clinic is a women's identity clinic.

This narrative describes how his need for anatomically-driven health care erases his gender identity due to the gendered environment. In addition, provider assumptions about what is causing the patient upset remain unexplored and thus worsen the situation.

Finally, clear and respectful provider communication skills were reported as critical to quality healthcare. A transgender woman (age 61) reported a good experience with her provider who explained clearly what would happen during her colonoscopy, including what would happen if they found a polyp:

> They informed me of every step that was going on.. . . If they did find a polyp, what would happen if they did find one. They were very informative of what was they doing to me, and what would the outcome be if it was, and if it were not.

A transgender woman (age 34) described her sense that providers did not want to answer her questions and felt uncomfortable:

> I mean, he could have listened to me. I never felt like they were listening. It was like they were just checking off different questions. "Do you feel this?" Click. "Do you feel this?" Click. I would say something and they'd say, "Oh, that's nothing." I don't feel like they were actually listening, you know? And he was a male anyway, so I always have this interesting relationship with males as my physicians or psychiatrists or something like that, because the emotional part seems to be, you can kind of see them cringing. Like, "Ugh, this is too emotional." Anyway, I'll just answer your questions and get out of here.

In this scenario, the provider's discomfort rushes the healthcare encounter, which the patient suggests has something to do with being a male physician. Another transgender woman (age 53) indicated that she has received very little education from her care team about cancer or required screening. When asked about what she can do to reduce her cancer risks, she said:

> I don't know. What can you do? I don't think you can do anything about it. I don't know. Really, I can't tell you, because I don't know where cancer comes from. I don't know if you can prevent it from eating. You can catch cancer from many ways. You see what I'm saying? You can catch cancer from many. . . from food. It can be from prostate. I don't know. You won't be able to tell.

In this case, the participant received no explanations at all about what types of things she could do to stay as healthy as possible.

One patient factor emerged as important for quality care: the ability to self-advocate. A transgender woman (age 34) said her health improvements came from her self-motivation:

> I changed my diet on my own and became vegan. So I was diabetic, but when I changed my diet I am no longer diabetic. So it was just like, "Oh, your numbers look great! Blah, blah, blah." I mean, that's good. But there are other things I've asked for that I need. . . [my doctor is] knowledgeable but she's the one that's not actually helping me do it. I'm really just doing it on my own. But I could use more support.

In this case, the participant respected her doctor's knowledge, but felt that she had to really own her own health outcomes. The above Emergency Department experience reported by a transgender male reinforced this theme for the need for patient self-advocacy.

A third overarching theme was the uniqueness of the TGD experience in healthcare. Unlike cisgender individuals, TGD people often experience gender dysphoria during cancer screening and other clinical procedures. In addition to gender dysphoria, other subthemes (child nodes) included managing provider gatekeeping in order to get health needs met, misgendering on a routine basis, fear of discrimination due to past negative healthcare experiences or trauma, and confusion regarding what cancer screenings to get and what procedures aligned with which cancer screenings. See Table 3 for quotations that illustrate these themes.

## Discussion

Results from this study suggest that there is great room for improvement in gender-affirming care as well as provision of cancer screening information for TGD people. Themes identified in this study support the need for broad training of health care providers to ensure professional, respectful health care for TGD people. Embedded in the dismissal of TGD patients by staff and clinicians is structural discrimination and stigma. Reflection on the pervasiveness of implicit bias and how it shapes health care delivery is critical to advance health equity for TGD people. Themes identified in this study also suggest additional training on interpersonal communication and empathy for front-line staff is needed. This finding supports prior qualitative work conducted with lesbian, gay, bisexual, and transgender, queer and intersex patients that

**Table 3. Unique experiences due to being transgender or gender nonconforming.**

| Theme | Quotation |
|---|---|
| Gender dysphoria | *I need them to be sensitive to my experience. And don't rush me, because my doctor kind of rushes me sometimes. I had something going on in the downstairs region I wanted her to look. . .It's always really awkward and I won't consider that disrespectful, but I know it's something that cisgender people don't really have to worry about when it comes to feeling awkward. I mean, maybe they do, but it's sometimes a really awkward feeling that I just ignore.* |
| | - Transgender woman, age 34 |
| Gender dysphoria specific to cancer screening | *With my level of dysphoria, I always knew I wanted all those parts gone. So I only would have forced myself through that examination with the outcome that it's going to be removed. So we're basically doing a final check before we remove it, you can do it, hang in there, we're gonna take this off. So for me I always felt like the uterus itself was the tumor. So for me the top parts themselves were the tumor.* |
| | - Transgender man, 45 |
| Managing gatekeeping | *I see this time and time again in the community, people skip an injection before their blood work, so it appears lower. . .So they can get their dose raised, because I kind of want and need to have levels that are above cisgender females. I'm not trying to simulate, I'm trying to transition.* |
| | - Transgender woman, age not reported |
| Misgendering | *They don't realize that treating me like an XX person is humiliating. It is absolutely, it's stripping me of my existence. I don't exist in that moment when you take that from me. I have no control. I mean, it's very similar to being raped. You're having your core identity stripped of you.* |
| | -Transgender man, age 45 |
| Fear of discrimination | *Well, being transgender, the thing that I worry about most is they're overall view of me because if they don't have a good overall view of me then they're not going to treat me the same as their other patients.* |
| | - Transgender woman, age 62 |
| Confusion about cancer screenings | *"I know the colon is five years. I don't know when. . . I think it's every five years. I don't know the turnaround period for the breast or the prostate."* |
| | - Transgender woman, age 61 |

identified the importance of respectful, collaborative partnerships between patients and physicians [32].

Findings from this study reinforce prior research exploring the needs of TGD patients. Lerner and Robles' 2017 review of extant research focused on TGD consumer perspectives of healthcare experiences noted that TGD patients frequently had negative healthcare encounters, often could not afford healthcare services, and faced gatekeeping among mental healthcare providers. In addition, providers frequently lacked knowledge of TGD issues and denied services, making it difficult for TGD patients to obtain necessary care [33]. Baldwin et al.'s 2018 study of TGD healthcare experiences found that TGD patients appreciated affirming language, experienced and knowledgeable providers, and the opportunity for gender identity disclosure [34]. Negative experiences included misgendering and outright transphobia. Our findings support this prior research. Specifically, the negative experiences recounted among our study participants and the ways in which TGD patients had to negotiate for quality care—including educating their providers and managing gatekeeping—are consistent with Lerner and Robles review. However, our study contributes to the literature by noting that gatekeeping occurs in a variety of medical scenarios, not just in mental healthcare encounters. In addition, themes of professionalism and clinical competence in TGD care that emerged in our study are consistent with Baldwin et al.'s findings.

A striking and novel finding in this study was the difference between health care experiences reported by transgender men versus transgender women. Transgender men reported extremely negative health care experiences and difficulty finding respectful, supportive providers. Transgender women in the sample reported greater satisfaction with their providers and more positive experiences with the health care system. The reasons for the different experiences among participants are not known. Based on references among transgender women in the study who self-identified as having HIV, this finding may be due, in part, to the frequency of health care needed for those who are living with HIV. In other words, those needing to access frequent health care may be more careful in choosing an affirming provider; however, this interpretation is speculative. More research is needed to identify differences between transgender men's and transgender women's health care experiences, trust/mistrust of health care providers and the health care system, and satisfaction with care. To our knowledge the differential experiences of transgender women versus transgender men in healthcare has not previously been reported in the literature.

A critical finding of this study was the impact of gender dysphoria on health care seeking behaviors. One participant said they went to the doctor "when the pain is so acute, or if I think I'm going to die" (Transgender male, age 45). Another participant described awkwardness in going to a doctor if there were any health problems involving her genitals (Transgender woman, age 34). Gender dysphoria was amplified by the gendered language of screening clinics and by misgendering and gatekeeping by clinicians. This finding has major implications for health and healthcare seeking among TGD individuals. Cervical cancer screening disparities among transgender men have been found in several prior studies [20, 35]. A recent review of barriers to cervical cancer screening among transgender men noted psychological and physical pain as deterrents to screening [36]. Another review supported these findings of lower cervical cancer screening uptake among transgender men using Behavioral Risk Factor Surveillance Survey data, and also noted significant gender identity disparities in colorectal cancer screening among transgender women [10]. Themes identified in the present study specific to respectful provider communication and affirming clinical environments are important strategies to reverse these trends.

Importantly, many participants in this study voiced confusion regarding which types of cancer screening were right for them. Many indicated that a provider never recommended a

cancer screening—not even colorectal cancer screening which should be universally recommended for anyone age 45 and over. This finding aligns with prior research [10] as well as the companion study of TGD individuals conducted in parallel with the present qualitative study: a survey of TGD individuals in the Washington, DC area (n = 58) found provider recommendations of cancer screening were statistically significantly associated with receipt of breast, prostate, colorectal, lung, skin, and anal screenings [37]. Misperceptions about which cancer screening procedures matched which body part being screened were pervasive. To our knowledge, our study is among the first to report on knowledge about cancer screenings among TGD people.

Collectively, our findings suggest a major failure of the health care delivery system at multiple levels for TGD patients. First, patients perceive that their providers lack knowledge, awareness and experience of their health care needs. These insufficiencies result in lack of cancer screening recommendations, missed opportunities to educate TGD about cancer risks and requisite screenings, and frequent occurrences of misgendering which discourage healthcare seeking and breeds mistrust among this patient population. Second, patients are not getting the health care information they need in a patient-friendly way to ensure optimal preventive care and early detection for cancer. Third, discrimination persists at the structural and provider levels. For example, body modification procedures not being performed, allowed, or covered by insurance at certain institutions systematically impedes access to gender affirming procedures that are critical to the psychosocial health and wellbeing of TGD people. Similarly, gender bias and discrimination at the provider level inhibit information sharing and education, lessen the likelihood that TGD patients will return for future appointments, and result in missed opportunities for preventive treatment and care.

While the authors acknowledge that each patient's lived experience is different and cancer may not be the top concern among many TGD people, it is paramount for healthcare providers to be giving TGD patients clear information on how to optimize their health, particularly for cancer screenings that are clearly evidence-based, such as cervical cancer screening for those with a cervix, and breast cancer and colorectal cancer screening based on recommended age and individual risk factors. Furthermore, these results suggest that patients may need repeated exposure to information to reinforce optimal health care utilization. Even individuals who had received cancer screenings, for example, did not always know the reason for the screening. No respondents voiced knowledge of optimal cancer screening intervals. These data support the need for clear communication by providers as well as access to comprehensive, coordinated care for patients.

Multi-disciplinary care can aid in more affirming care and clearer explanations of care. Psychologists, for example, are well poised to assist with improved communication between providers and patients. Within the Veterans Health Administration, psychologists often serve on interdisciplinary teams in an effort to ensure TGD veterans understand risks and benefits of medical and mental health interventions as well as to coordinate care among providers [38]. Integrated care teams are uniquely positioned to positively impact individuals at risk for adverse health outcomes [39], and psychologists with appropriate specialty backgrounds serve roles in shaping a culturally responsive environment and increasing health literacy in patient-provider communications among others [40].

## Limitations

This study has several important limitations. First, there were more transgender women compared to transgender men in the study. While the study team tried to achieve a balanced sample, recruitment of transgender men was much more difficult. The reasons for this are

unknown, but it appeared, based on anecdotes from participants, that transgender women were closely connected and referred to the study through word of mouth whereas transgender men were individually recruited in response to tabling at community events or an announcement by a TGD-serving community organization. In addition, the self-disclosure of HIV status among many transgender women in the study could mean that the transgender female sample does not reflect the general experiences of transgender women, including those who are not living with HIV. As a qualitative study, participant perceptions of social desirability are also always a potential bias.

## Trustworthiness

Credibility of this study was established through the use of two coders with qualitative research expertise, as well as member checking. The two coders (MPC and JM) met regularly to discuss emergent themes, establish a codebook, and refine examples illustrating themes. Member checking was conducted by inviting all interviewees to attend a private webinar discussing results of the study to ensure that nothing was misinterpreted or lacking thematically. Three interviewees chose to attend the webinar and provided positive feedback on interpretation of findings. A third member of the research team (DH) reviewed congruence of findings with her experience interviewing participants. Additionally, team members reviewed findings from multi-disciplinary clinical perspectives (AR, DH, AW).

## Conclusions

This study suggests that there is significant room for improvement in the provision of health care information to TGD individuals to optimize their health literacy, specifically in the area of cancer screening. Results also suggest a need for improvements to provider communication skills, clinical knowledge, and cultural competency through training and education. Finally, respondents supported the need for improved care coordination and insurance navigation among all TGD people. These study findings are strengthened by findings in a parallel quantitative study, which found that provider recommendations were associated with receipt of cancer screenings in a TGD sample (n = 58) and that interpersonal skills, affirming language, and clear information were provider characteristics valued by TGD patients [37]. More research is warranted to identify effective interventions to address these challenges.

## Acknowledgments

The authors would like to thank the following individuals for their thoughtful feedback on this manuscript: Tony Burns, Alayna Waldrum, Aundra Campbell, and Derrick Cox. The authors would also like to thank the DC Area Transmasculine Society for assisting with recruitment for the study and the GW Cancer Center LGBQI Community Advisory Board for their support throughout study conceptualization, implementation, and analysis.

## Author Contributions

**Conceptualization:** Mandi L. Pratt-Chapman.

**Data curation:** Mandi L. Pratt-Chapman, Dana Hines, Ruta Brazinskaite.

**Formal analysis:** Mandi L. Pratt-Chapman, Jeanne Murphy.

**Funding acquisition:** Mandi L. Pratt-Chapman.

**Investigation:** Mandi L. Pratt-Chapman.

**Methodology:** Mandi L. Pratt-Chapman.

**Project administration:** Mandi L. Pratt-Chapman.

**Supervision:** Mandi L. Pratt-Chapman.

**Writing – original draft:** Mandi L. Pratt-Chapman, Jeanne Murphy, Ruta Brazinskaite.

**Writing – review & editing:** Mandi L. Pratt-Chapman, Dana Hines, Allison R. Warren, Asa Radix.

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
