## [Decision Letter · Decision Letter 0]

17 Nov 2020

PONE-D-20-32209

“When the Pain is so Acute or if I think that I’m going to Die”: Health Care Seeking Behaviors and Experiences of Transgender and Gender Diverse People in an Urban Area

PLOS ONE

Dear Dr. Chapman,

Thank you for submitting your manuscript to PLOS ONE. After careful consideration, we feel that it has merit but does not fully meet PLOS ONE’s publication criteria as it currently stands. Therefore, we invite you to submit a revised version of the manuscript that addresses the points raised during the review process.

Please see comments from reviewers and make the minor edits to the manuscript (particularly the suggested table and discussion items). We look forward to receiving your revised manuscript for publication.

We look forward to receiving your revised manuscript.

Kind regards,

Amy Michelle DeBaets, PhD

Academic Editor

PLOS ONE

Journal Requirements:

2. We note you have included a table to which you do not refer in the text of your manuscript. Please ensure that you refer to Table 1 in your text; if accepted, production will need this reference to link the reader to the Table.

3.We note that you have indicated that data from this study are available upon request. PLOS only allows data to be available upon request if there are legal or ethical restrictions on sharing data publicly. For information on unacceptable data access restrictions, please see http://journals.plos.org/plosone/s/data-availability#loc-unacceptable-data-access-restrictions.

Reviewers' comments:

Reviewer's Responses to Questions

**Comments to the Author**

1. Is the manuscript technically sound, and do the data support the conclusions?

Reviewer #1: Yes

Reviewer #2: Yes

2. Has the statistical analysis been performed appropriately and rigorously? 

Reviewer #1: N/A

Reviewer #2: N/A

3. Have the authors made all data underlying the findings in their manuscript fully available?

Reviewer #1: Yes

Reviewer #2: Yes

4. Is the manuscript presented in an intelligible fashion and written in standard English?

Reviewer #1: Yes

Reviewer #2: Yes

5. Review Comments to the Author

Reviewer #1: This study explored the general healthcare experiences of transgender and diverse (TGD) people in the Washington, DC area, and cancer screening experiences in particular. Results suggested a need for improved provider communication skills, including clear explanations of procedures and recommendations for appropriate screenings to TGD patients. Results also suggested a need for improved clinical knowledge and cultural competency. The study provides very important information, is methodologically sound and the paper well written. The study however is limited by small sample size, patient selection bias, self reporting of medical history and qualitative design.

Minor points

A summary table is suggested.

more discussion is needed on how this study compares with earlier literature

Reviewer #2: Thanks for choosing to work on the marginalized community. I find that using both transmasculine/feminine and transgender man/feminine terms inter-changeably may be confusing for the audience not familiar with the transgender population literature and terminology.

It would enrich the manuscript to first do a table with all the themes that emerged, then detail thick description of the themes with the excepts from the participants. Further under discussion, it would be good to contrast the findings of this study, with what is out there in terms of literature that supports or disputes your findings.

6. PLOS authors have the option to publish the peer review history of their article (what does this mean?). If published, this will include your full peer review and any attached files.

Reviewer #1: No

Reviewer #2: **Yes: **Dr Zamasomi Luvuno

---

## [Author Response · Author response to Decision Letter 0]

11 Dec 2020

Response to Reviewers

Thank you. We have revised based on your style guide. 

2. We note you have included a table to which you do not refer in the text of your manuscript. Please ensure that you refer to Table 1 in your text; if accepted, production will need this reference to link the reader to the Table.

Table 1 is now referred to in the text.

We sought guidance from our Office of Human Research (OHR). Because our informed consent indicated that only the study team would have access to transcripts and due to the sensitive nature of the interviews, the OHR indicated that transcripts should not be shared. 

In your revised cover letter, please address the following prompts: a) If there are ethical or legal restrictions on sharing a de-identified data set, please explain them in detail (e.g., data contain potentially identifying or sensitive patient information) and who has imposed them (e.g., an ethics committee). Please also provide contact information for a data access committee, ethics committee, or other institutional body to which data requests may be sent.

We have provided a revised cover letter indicating the IRB’s concerns. 

Thank you.

A summary table is suggested.

A summary table has been provided (Table 2 on pages 7-8 of the clean manuscript). The prior long table was removed and examples incorporated into the Results section. The deletion of Table 2 does not show up in tracked changes, but the modification is apparent in the additional narrative in the Results section (pages 8-15). Table 3 (misnumbered previously as Table 4, now page 16) was retained to succinctly illustrate the themes unique to TGD experience. We felt this short table was easier for a reader than the long table of results that has now been incorporated into the narrative of the results section.

More discussion is needed on how this study compares with earlier literature.

Additional context for how the study compares to prior literature has been added to the discussion (pages 18-22).

Reviewer #2: Thanks for choosing to work on the marginalized community. I find that using both transmasculine/feminine and transgender man/feminine terms inter-changeably may be confusing for the audience not familiar with the transgender population literature and terminology.

Thank you for this feedback. The original language was intentional in order to indicate that trans experience occurs over a wide spectrum. However, we have changed the terminology to eliminate confusion with the exception of “Transmasculine” in the name of a community organization, since that is a proper noun. 

It would enrich the manuscript to first do a table with all the themes that emerged, then detail thick description of the themes with the excepts from the participants. Further under discussion, it would be good to contrast the findings of this study, with what is out there in terms of literature that supports or disputes your findings.

A summary table has been provided (Table 2). Other tables have been removed and examples incorporated into the Results and Discussion sections. Additional context for how the study compares to prior literature has been added to the discussion (pages 20-22).

---

## [Decision Letter · Decision Letter 1]

28 Jan 2021

“When the Pain is so Acute or if I think that I’m going to Die”: Health Care Seeking Behaviors and Experiences of Transgender and Gender Diverse People in an Urban Area

PONE-D-20-32209R1

Dear Dr. Chapman,

We’re pleased to inform you that your manuscript has been judged scientifically suitable for publication and will be formally accepted for publication once it meets all outstanding technical requirements.

Kind regards,

Amy Michelle DeBaets, PhD

Academic Editor

PLOS ONE

Additional Editor Comments (optional):

Reviewers' comments:

Reviewer's Responses to Questions

**Comments to the Author**

1. If the authors have adequately addressed your comments raised in a previous round of review and you feel that this manuscript is now acceptable for publication, you may indicate that here to bypass the “Comments to the Author” section, enter your conflict of interest statement in the “Confidential to Editor” section, and submit your "Accept" recommendation.

Reviewer #1: All comments have been addressed

2. Is the manuscript technically sound, and do the data support the conclusions?

Reviewer #1: Yes

3. Has the statistical analysis been performed appropriately and rigorously? 

Reviewer #1: N/A

4. Have the authors made all data underlying the findings in their manuscript fully available?

Reviewer #1: Yes

5. Is the manuscript presented in an intelligible fashion and written in standard English?

Reviewer #1: Yes

6. Review Comments to the Author

Reviewer #1: (No Response)

7. PLOS authors have the option to publish the peer review history of their article (what does this mean?). If published, this will include your full peer review and any attached files.

Reviewer #1: No

---

## [Editor Report · Acceptance letter]

9 Feb 2021

PONE-D-20-32209R1 

“When the pain is so acute or if I think that I’m going to die”: Health care seeking behaviors and experiences of transgender and gender diverse people in an urban area 

Dear Dr. Pratt-Chapman:

I'm pleased to inform you that your manuscript has been deemed suitable for publication in PLOS ONE. Congratulations! Your manuscript is now with our production department. 

Kind regards, 

on behalf of

Dr. Amy Michelle DeBaets 

Academic Editor

PLOS ONE